# CROSS-SILO TRAINING OF DIFFERENTIALLY PRIVATE MODELS WITH SECURE MULTIPARTY COMPUTATION

## ABSTRACT

We address the problem of learning a machine learning model from training data that originates at multiple data owners in a cross-silo federated setup, while providing formal privacy guarantees regarding the protection of each owner's data. Existing solutions based on Differential Privacy (DP) achieve this at the cost of a drop in accuracy. Solutions based on Secure Multiparty Computation (MPC) do not incur such accuracy loss but leak information when the trained model is made publicly available. We propose an MPC solution for training differentially private models. Our solution relies on an MPC protocol for model training, and an MPC protocol for perturbing the trained model coefficients with Laplace noise in a privacy-preserving manner. The resulting MPC+DP approach achieves higher accuracy than a pure DP approach, while providing the same formal privacy guarantees.

## 1 INTRODUCTION

The ability to induce a machine learning (ML) model from data that originates at multiple data owners (clients) in a cross-silo federated setup, while protecting the privacy of each data owner, is of great practical value in a wide range of applications, for a variety of reasons. Most prominently, training on more data typically yields higher quality ML models. For instance, one could train a more accurate model to predict the length of hospital stay of COVID-19 patients when combining data from multiple clinics. This is a cross-silo application where the data is *horizontally distributed*, meaning that each data owner (clinic) has records/rows of the data (HFL). Furthermore, being able to combine different data sets enables new applications that pool together data from multiple data owners, or even from different data owners within the same organization. An example of this is an ML model that relies on lab test results as well as healthcare bill payment information about patients, which are usually managed by different departments within a hospital system. This is an example of a cross-silo application where the data is *vertically distributed*, i.e. each data owner has their own columns (VFL). While there are clear advantages to training ML models over data that is distributed across multiple data owners, in practice often these data owners do not want to disclose their data to each other, because the data in itself constitutes a competitive advantage, or because the data owners need to comply with data privacy regulations.

The importance of enabling privacy-preserving model training in federated setups has spurred a large research effort in this domain, most notably in the development and use of Privacy-Enhancing Technologies (PETs), prominently including Federated Learning (FL) (Kairouz et al. (2021)), Differential Privacy (DP) (Dwork et al. (2014)), Secure Multiparty Computation (MPC) (Cramer et al. (2015)), and Homomorphic Encryption (HE) (Lauter (2021)). Each of these techniques has its own (dis)advantages. Approaches based on (combinations of) FL, MPC, or HE alone do not provide sufficient protection if the trained model is to be made publicly known, or even if it is only made available for black-box query access, because information about the model and its training data is leaked through the ability to query the model (Fredrikson et al. (2015); Tramèr et al. (2016); Song et al. (2017); Carlini et al. (2019)). Formal privacy guarantees in this case can be provided by DP, however at a cost of accuracy loss that is inversely proportional to the *privacy budget* (see Sec. 2). To mitigate this accuracy loss, we propose an MPC solution for training DP models.

**Our Approach.** Rather than having each party training local models on their own data sets, we have the parties running an MPC protocol on the totality of the data sets without requiring each party to disclose their private information to anyone. Since we restrict our analysis to generalized linear models, we then have these parties using MPC to generate the necessary noise and privately adding

it to the weights of the trained classifier to satisfy DP requirements. We show that this procedure yields the same accuracy and DP guarantees as in the global DP model however without requiring the parties to reveal their data, model parameters, or gradients to a central aggregator, or to anyone else for that matter. Indeed, the MPC protocols effectively play the role of a trusted curator implementing global DP. The resulting classifier can be published in the clear, or used for private inference on top of MPC. Our solution is applicable in cross-silo federated scenarios in which the data is horizontally distributed as well as in cross-silo federated scenarios where the data is vertically distributed. It obtained the highest accuracy in the iDASH2021 Track III competition on confidential computing, where the challenge was to propose a federated learning algorithm for training of a model to predict the risk of wild-type transthyretin amyloid cardiomyopathy using medical claims data from different hospitals, while providing DP guarantees.[1]

## 2 PRELIMINARIES

**Differential Privacy.** DP is concerned with providing aggregate information about a data set $D$ without disclosing information about specific individuals in $D$ (Dwork et al. (2014)). A data set $D'$ that differs in a single entry from $D$ is called a neighboring database. A randomized algorithm $\mathcal{A}$ is called $(\epsilon, \delta)$-DP if for all pairs of neighboring databases $D$ and $D'$, and for all subsets $S$ of $\mathcal{A}$'s range,

$$\mathrm{P}(\mathcal{A}(D) \in S) \leq e^{\epsilon} \cdot \mathrm{P}(\mathcal{A}(D') \in S) + \delta. \tag{1}$$

In other words, $\mathcal{A}$ is DP if $\mathcal{A}$ generates similar probability distributions over outputs on neighboring data sets $D$ and $D'$. The parameter $\epsilon \geq 0$ denotes the *privacy budget* or privacy loss, while $\delta \geq 0$ denotes the probability of violation of privacy, with smaller values indicating stronger privacy guarantees in both cases. $\epsilon$-DP is a shorthand for $(\epsilon, 0)$-DP. $\mathcal{A}$ can for instance be an algorithm that takes as input a data set $D$ of training examples and outputs an ML model. An $(\epsilon, \delta)$-DP randomized algorithm $\mathcal{A}$ is commonly created out of an algorithm $\mathcal{A}^*$ by adding noise that is proportional to the sensitivity of $\mathcal{A}^*$. We describe the Laplace noise technique that we use to this end in detail in Sec. 4.

**Secure Multiparty Computation.** MPC is an umbrella term for cryptographic approaches that allow two or more parties to jointly compute a specified output from their private information in a distributed fashion, without revealing the private information to each other (Cramer et al. (2015)). MPC is concerned with the protocol execution coming under attack by an adversary which may corrupt one or more of the parties to learn private information or cause the result of the computation to be incorrect. MPC protocols are designed to prevent such attacks being successful, and can be mathematically proven to guarantee privacy and correctness. We follow the standard definition of the Universal Composability (UC) framework (Canetti (2000)), in which the security of protocols is analyzed by comparing a real world with an ideal world. For details, see Evans et al. (2018).

An adversary can corrupt a certain number of parties. In a *dishonest-majority* setting the adversary is able to corrupt half of the parties or more if he wants, while in an *honest-majority* setting, more than half of the parties are always honest (not corrupted). Furthermore, the adversary can have different levels of adversarial power. In the *semi-honest* model, even corrupted parties follow the instructions of the protocol, but the adversary attempts to learn private information from the internal state of the corrupted parties and the messages that they receive. MPC protocols that are secure against semi-honest or *"passive"* adversaries prevent such leakage of information. In the *malicious* adversarial model, the corrupted parties can arbitrarily deviate from the protocol specification. Providing security in the presence of malicious or *"active"* adversaries, i.e. ensuring that no such adversarial attack can succeed, comes at a higher computational cost than in the passive case. The protocols in Sec. 4 are sufficiently generic to be used in dishonest-majority as well as honest-majority settings, with passive or active adversaries. This is achieved by changing the underlying MPC scheme to align with the desired security setting.

As an illustration, we describe the well-known additive secret-sharing scheme for dishonest-majority 2PC with passive adversaries. In Sec. 5 we additionally present results for honest-majority 3PC and 4PC schemes with passive and active adversaries; for details about those MPC schemes we refer to Araki et al. (2016); Dalskov et al. (2021). In the additive secret-sharing 2PC scheme there are two computing parties, nicknamed *Alice* and *Bob*. All computations are done on integers, modulo an integer $q$. The modulo $q$ is a hyperparameter that defines the algebraic structure in which the computations are done. A value $x$ in $\mathbb{Z}_q = \{0, 1, \ldots, q-1\}$ is secret shared between Alice and Bob

---

[1]http://www.humangenomeprivacy.org/2021/competition-tasks.html

by picking uniformly random values $x_1, x_2 \in \mathbb{Z}_q$ such that $x_1 + x_2 = x \mod q$. $x_1$ and $x_2$ are additive shares of $x$ (which are delivered to Alice and Bob, respectively). Note that no information about the secret value $x$ is recovered by any of the individual shares $x_1$ or $x_2$, but the secret-shared value $x$ can be trivially revealed by combining both shares $x_1$ and $x_2$. The parties Alice and Bob can jointly perform computations on numbers by performing computations on their own shares, without the parties learning the values of the numbers themselves.

For protocols in the passive-security setting, we use $[\![x]\!]$ as a shorthand for a secret sharing of $x$, i.e. $[\![x]\!] = (x_1, x_2)$. Given secret-shared values $[\![x]\!] = (x_1, x_2)$ and $[\![y]\!] = (y_1, y_2)$, and a constant $c$, Alice and Bob can jointly perform the following operations, each by doing only local computations on their own shares:[2]

- Addition of a constant ($z = x + c$): Alice and Bob compute $(x_1 + c, x_2)$. Note that Alice adds $c$ to her share $x_1$, while Bob keeps the same share $x_2$. This operation is denoted by $[\![z]\!] \leftarrow [\![x]\!] + c$.
- Addition ($z = x + y$): Alice and Bob compute $(x_1 + y_1, x_2 + y_2)$ by adding their local shares of $x$ and $y$. This operation is denoted by $[\![z]\!] \leftarrow [\![x]\!] + [\![y]\!]$.
- Multiplication by a constant ($z = c \cdot x$): Alice and Bob compute $(c \cdot x_1, c \cdot x_2)$ by multiplying their local shares of $x$ by $c$. This operation is denoted by $[\![z]\!] \leftarrow c[\![x]\!]$.

Multiplication of secret-shared values $[\![x]\!]$ and $[\![y]\!]$ is done using a so-called *multiplication triple* (Beaver (1992)), which is a triple of secret-shared values $[\![u]\!]$, $[\![v]\!]$, $[\![w]\!]$, such that $u$ and $v$ are uniformly random values in $\mathbb{Z}_q$ and $w = u \cdot v$. Given that they have a multiplication triple, Alice and Bob can compute $[\![d]\!] = [\![x]\!] - [\![u]\!]$ and $[\![e]\!] = [\![y]\!] - [\![v]\!]$, and, in a communication step, *open* $d$ and $e$ by disclosing their respective shares of $d$ and $e$ to each other. Next, they can compute $[\![z]\!] = [\![w]\!] + d \cdot [\![v]\!] + e \cdot [\![u]\!] + d \cdot e$, which is equal to $[\![x \cdot y]\!]$. We denote this operation by $[\![z]\!] \leftarrow \pi_{\mathsf{MUL}}([\![x]\!], [\![y]\!])$. Each multiplication requires a fresh multiplication triple. Such triples can be predistributed by a trusted initializer (TI). In case a TI is not available or desirable, Alice and Bob can simulate the role of the TI, at the cost of additional pre-processing time and computational assumptions, see Mohassel and Zhang (2017).

Building on these cryptographic primitives, MPC protocols for other operations can be developed, including for privacy-preserving training of ML models and noise generation to provide DP guarantees (see Sec. 4). Our protocols use well known subprotocols for division $\pi_{\mathsf{DIV}}$ of secret-shared values, square root $\pi_{\mathsf{SQRT}}$ of secret-shared values, and generation of random values from a uniform distribution $\pi_{\mathsf{GR-RANDOM}}$ (Keller (2020)).

## 3 RELATED WORK

Our approach preserves input privacy, i.e., it ensures that the training data sets are not exposed (except under $\epsilon$-DP guarantees) to anyone but their original holders during (1) model training and (2) publication or inference. As we describe below, existing methods either do not fully protect input privacy, or they do so at the cost of higher accuracy loss than our approach.

**MPC/HE based Model Training.** Many cryptography based methods have been proposed for privacy-preserving learning of ML models with data from multiple data owners, including for linear regression models (Gascón et al. (2017); Agarwal et al. (2019)), (ensembles of) decision trees (Lindell and Pinkas (2000); de Hoogh et al. (2014); Abspoel et al. (2021); Adams et al. (2022)), and neural network architectures (Mohassel and Zhang (2017); Wagh et al. (2019); Guo et al. (2020); De Cock et al. (2021)). These techniques protect input privacy during training while still, in principle, producing the same ML models that one would obtain in the clear (i.e. when no encryption is used). The latter is both a blessing, as there is no accuracy loss, and a problem, as upon model publication or during inference, the trained models leak the same kind of information as models trained in-the-clear (Fredrikson et al. (2015); Tramèr et al. (2016); Song et al. (2017); Carlini et al. (2019)). Because these methods do not provide DP guarantees, we do not compare with them in Sec. 5.

**DP and FL based Model Training.** Much of the literature on training DP models (Abadi et al. (2016)) is developed for the *global* DP (a.k.a. *central* DP) paradigm, which assumes the existence of a trusted curator (aggregator) who collects all the data and then trains a DP model over it, e.g. by adding noise to the gradients or the model coefficients. These methods do not preserve input privacy, since data owners need to disclose their data sets to the aggregator. A *local* DP approach in which privacy loss is controlled by having the data owners add noise to their input data *before* disclosing it

---

[2]We often omit the modular notation for conciseness.

to the aggregator, results in substantial utility degradation. We eliminate the need for a trusted curator by simulating this entity through MPC protocols that are run directly by the parties themselves.

Another related existing approach combines Federated Learning (FL) with DP. In FL, each of the data owners participates in model training on their end and only exchanges trained model parameters or gradients with the central server (Kairouz et al. (2021)). To provide DP guarantees, the data owners can add noise to protect the values that they send to the central server. In Sec. 5 we compare with such an approach in which the data owners perturb their model coefficients before sending them to the central server for aggregation. This approach works only in the horizontally distributed data setting, while our approach (see Sec. 4) works in the vertically distributed setting as well.

**Combinations of MPC and DP.** The key idea in our proposed approach is to train DP models while performing as much of the computations as possible in MPC protocols in order to preserve accuracy. MPC and DP for ML have been well studied in isolation, but the strong privacy protections that can result from their synergy are still being explored (Wagh et al. (2021)). We combine MPC and DP to protect training data privacy during training *and* during inference. In practice, we simulate the trusted curator present in the centralized DP model by using MPC. While in the past such approach was avoided, due to the high computational cost of training the models on top of MPC, we argue that, with advances in protocols and computing power, the higher utility provided by such approach justifies its adoption in several situations. The idea to replace the trusted curator from the global DP paradigm with MPC to get better privacy at the same high utility will gain traction. Böhler and Kerschbaum (2021) for instance have recently explored this idea for detecting the top $k$ most frequent items across different data sets. They let each party locally compute partial noises which are then combined, which is different from our approach of letting the parties execute an MPC protocol to jointly sample secret-shared noise.

Combining MPC with DP has been proposed in the context of FL, where the data is *horizontally* distributed (see e.g. Acar et al. (2017); Jayaraman et al. (2018); Pathak et al. (2010)). No solutions for the *vertically* partitioned scenario exist. Another possible approach is to use cryptographic protocols (not necessarily MPC) and differential privacy, such as in Jayaraman et al. (2018); Pathak et al. (2010); Chase et al. (2017); Byrd and Polychroniadou (2020); Truex et al. (2019), in order to train individual models on the data sets in possession of the computing parties and aggregate these models by averaging their coefficients. Again, this approach does not work for vertically partitioned data. Moreover, our solution trains the final model on the union of all the individual data sets, thus essentially obtaining the same utility that is achievable in the trusted curator scenario. Protocols for obliviously sampling from biased coins on MPC have been proposed in Champion et al. (2019). In Gu et al. (2021), a framework for combining MPC and Federated learning is proposed, but it only works for the case of horizontally partitioned data. Moreover, the noise generation process described in Gu et al. (2021) happens in the clear (by each server) and it is combined by using secret sharing. That requires more noise than in our proposal (where the noise is generated within MPC), thus reducing the utility of the data.

## 4 METHOD

**Overview.** We work in the scenario described in Fig. 1 distinguishing between the *data owners* who hold the training data sets, and the *computing parties* who run the MPC protocols for model training and noise addition. Our solution works in scenarios in which each data owner (e.g. hospital or bank) is also a computing party, as well as in scenarios where the data owners outsource the computations to untrusted servers (computing parties) instead. The data holders secret share their data with a set of computing servers. The servers run an MPC protocol and produce an ML model protected by DP. We implement our solution for 2, 3 and 4 computing servers, but they are general and work with any number of computing servers as well as data holders, by choosing an appropriate underlying MPC scheme for the desired number of computing parties (see Sec. 2). The resulting model can be used for private inference (on top of the underlying MPC protocol) or made open to the public.

The core of our solution is an MPC protocol $\pi_{\mathsf{DP}}$ implementing a mechanism for providing $\epsilon$-DP by perturbing the coefficients of a trained logistic regression (LR) model with the addition of a noise vector $\eta$ that is sampled according to the density function

$$h(\eta) \propto e^{-\frac{n\epsilon\Lambda}{2}\|\eta\|} \tag{2}$$

In the above expression, $n$ is the number of instances that were used to train the LR model, and $\Lambda$ is the regularization strength parameter used during training. This technique provides $\epsilon$-DP provided

that (C1) each input feature vector has an L2 norm of at most 1; and (C2) the LR model is trained using L2 regularization. If (C1) and (C2) are fulfilled, then the sensitivity of LR with regularization parameter $\Lambda$ is at most $\frac{2}{n\Lambda}$ (Chaudhuri and Monteleoni (2008); Chaudhuri et al. (2011)).

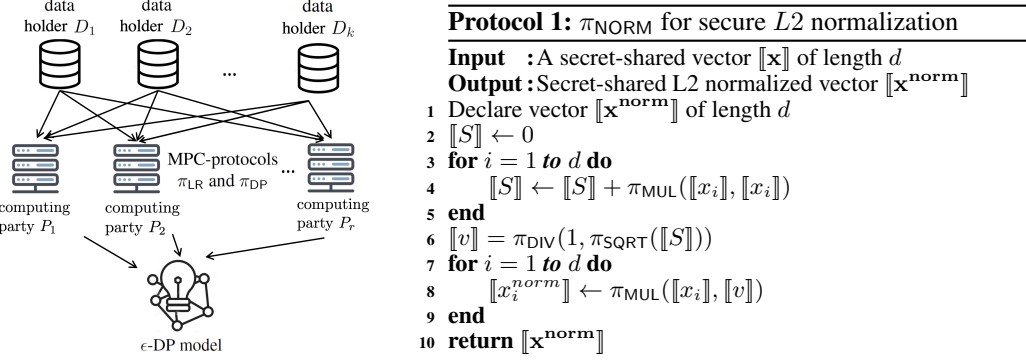

**Protocol 1:** $\pi_{\mathsf{NORM}}$ for secure $L2$ normalization

**Input** : A secret-shared vector $[\![\mathbf{x}]\!]$ of length $d$
**Output** : Secret-shared L2 normalized vector $[\![\mathbf{x^{norm}}]\!]$
1 Declare vector $[\![\mathbf{x^{norm}}]\!]$ of length $d$
2 $[\![S]\!] \leftarrow 0$
3 **for** $i = 1$ **to** $d$ **do**
4     $[\![S]\!] \leftarrow [\![S]\!] + \pi_{\mathsf{MUL}}([\![x_i]\!], [\![x_i]\!])$
5 **end**
6 $[\![v]\!] = \pi_{\mathsf{DIV}}(1, \pi_{\mathsf{SQRT}}([\![S]\!]))$
7 **for** $i = 1$ **to** $d$ **do**
8     $[\![x_i^{norm}]\!] \leftarrow \pi_{\mathsf{MUL}}([\![x_i]\!], [\![v]\!])$
9 **end**
10 **return** $[\![\mathbf{x^{norm}}]\!]$

Figure 1: Privacy-preserving training of an $\epsilon$-DP model with MPC

In all MPC protocols used in this paper, secret sharings are in $\mathbb{Z}_q$ with $q = 2^\lambda$, i.e. a power 2 ring. In Sec. 5 we present results with $\lambda = 64$ for a varying number of data owners, and for 2, 3, and 4 computing parties. Since all computations in MPC are done over integers in $\mathbb{Z}_q$ (see Sec. 2), the data owners first convert the real numbers in their data to integers using a fixed-point representation (Catrina and Saxena (2010)) and subsequently split the integer values into secret shares which are sent to the computing parties (see Fig. 1). While the original value of a secret-shared number can be trivially revealed by combining the shares, the secret-sharing based MPC schemes ensure that nothing about the inputs is revealed to any subset of the computing parties that can be corrupted by an adversary. This means, in particular, that no computing party by itself learns anything about the actual values of the inputs. Next, the computing parties proceed by performing computations on the shares. In particular, the computing parties:

1. Jointly run MPC protocol $\pi_{\mathsf{LR}}$ to L2 normalize the training data, and to subsequently infer a LR model using L2 regularization from the normalized data. At the end of this protocol, the coefficients of the model are secret-shared between the parties.
2. Jointly run MPC protocol $\pi_{\mathsf{DP}}$ to add a noise vector to the secret-shared model coefficients. At the end of this protocol, the noisy coefficients of the model are secret shared between the parties.
3. Disclose their shares of the LR coefficients so that they can be combined in a final $\epsilon$-DP LR model.

As the noise in step 2 is generated and added to the weights using MPC, the computing parties will not learn it, hence they will not be able to retrieve the actual model coefficients from the noisy coefficients that are disclosed in step 3.

**Protocol $\pi_{\mathsf{LR}}$ for Model Training.** At the beginning of the LR training protocol, the computing parties have secret shares of a set of labeled training examples $S = \{([\![\mathbf{x}]\!], [\![t]\!])\}$, each consisting of a secret-shared input feature vector $\mathbf{x}$ of length $m$ and a secret shared label $t$. $\pi_{\mathsf{LR}}$ is based on an existing MPC protocol for training a LR with full gradient descent (GD) (Keller (2020)). We extended this protocol in two ways. First, to satisfy condition (C1), before the start of model training, we let the computing parties apply L2 normalization to the secret shares of each training example $[\![\mathbf{x^{norm}}]\!]$ by running $\pi_{\mathsf{NORM}}$. Pseudocode for $\pi_{\mathsf{NORM}}$ is provided separately in Prot. 1 because we also need it as a subprotocol for $\pi_{\mathsf{DP}}$. If the data is horizontally distributed across the data owners, then each data owner can apply sample-wise L2 normalization to their own instances before secret sharing the training instances with the computing parties. The computing parties in this case can skip the use of $\pi_{\mathsf{NORM}}$ for this purpose, which will reduce the training runtime. Second, to comply with condition (C2), we implemented regularization by changing the weight update rule to $[\![\Delta\mathbf{w}]\!] \leftarrow C[\![\Delta\mathbf{w}]\!] - \alpha[\![\Delta\mathbf{w}]\!] - \Lambda\alpha[\![\mathbf{w}]\!]$. In this expression, $[\![\mathbf{w}]\!]$ and $[\![\Delta\mathbf{w}]\!]$ are the weights and gradients as maintained in secret-shared form throughout the model training; $C$ is the momentum; $\alpha$ is the learning rate; and $\Lambda$ is the regularization penalty. Pseudocode for $\pi_{\mathsf{LR}}$ is provided in Appendix A.

**Protocol 2:** $\pi_{\mathsf{DP}}$ for secure output perturbation

| | |
|---|---|
| **Input** | : A secret-shared vector $[\![\mathbf{w}]\!]$ with $d$ model coefficients $w_i$; regularization penalty $\Lambda$; total number $n$ of training examples; privacy budget $\epsilon$. |
| **Output** | : Secret-shared vector $[\![\widetilde{\mathbf{w}}]\!]$ with perturbed model coefficients |

1  $[\![\mathbf{s}]\!] \leftarrow \pi_{\mathsf{GSS}}(d)$
2  $[\![\mathbf{s}]\!] \leftarrow \pi_{\mathsf{NORM}}([\![\mathbf{s}]\!], d)$
3  $[\![\gamma]\!] \leftarrow [\![0]\!]$
4  **for** $i = 1$ **to** $d$ **do**
5      $[\![u]\!] \leftarrow \pi_{\mathsf{GR-RANDOM}}(0, 1)$
6      $[\![u]\!] \leftarrow -\pi_{\mathsf{LN}}([\![u]\!])$
7      $[\![\gamma]\!] \leftarrow [\![\gamma]\!] + [\![u]\!]$
8  **end**
9  $c \leftarrow 2/(n \cdot \epsilon \cdot \Lambda)$
10 $[\![\gamma]\!] \leftarrow c[\![\gamma]\!]$
11 Initialize vector $[\![\widetilde{\mathbf{w}}]\!]$ of length $d$ to $[\![\mathbf{0}]\!]$
12 **for** $i = 1$ **to** $d$ **do**
13     $[\![s_i]\!] \leftarrow \pi_{\mathsf{MUL}}([\![s_i]\!], [\![\gamma]\!])$
14     $[\![\widetilde{w}_i]\!] \leftarrow [\![w_i]\!] + [\![s_i]\!]$
15 **end**
16 **return** $[\![\widetilde{\mathbf{w}}]\!]$

**Protocol 3:** $\pi_{\mathsf{GSS}}$ for secure sampling of a vector from a Gaussian distribution

| | |
|---|---|
| **Input** | : Vector length $d$. |
| **Output** | : A secret-shared vector $[\![\mathbf{s}]\!]$ of length $d$ sampled from Gaussian distribution with mean 0 and variance 1 |

1  Declare vector $[\![\mathbf{s}]\!]$ of length $d$
2  **for** $i = 0$ **to** $d/2$ **do**
3      $[\![u]\!] \leftarrow \pi_{\mathsf{GR-RANDOM}}(0, 1)$
4      $[\![v]\!] \leftarrow \pi_{\mathsf{GR-RANDOM}}(0, 1)$
5      $[\![r]\!] \leftarrow \pi_{\mathsf{SQRT}}(-2\pi_{\mathsf{LN}}([\![u]\!]))$
6      $[\![\theta]\!] \leftarrow 2\pi[\![v]\!]$
7      $[\![s_{2i}]\!] \leftarrow \pi_{\mathsf{MUL}}([\![r]\!], \pi_{\mathsf{COS}}([\![\theta]\!]))$
8      $[\![x_{2i+1}]\!] \leftarrow \pi_{\mathsf{MUL}}([\![r]\!], \pi_{\mathsf{SIN}}([\![\theta]\!]))$
9  **end**
10 **if** $d$ *is odd* **then**
11     $[\![p]\!] \leftarrow \pi_{\mathsf{GSS}}(2)$
12     $[\![s_{d-1}]\!] \leftarrow [\![p_0]\!]$
13 **end**
14 **return** $[\![\mathbf{s}]\!]$

**Protocol $\pi_{\mathsf{DP}}$ for Noise Generation.** At the end of MPC protocol $\pi_{\mathsf{LR}}$, the coefficients $\mathbf{w}$ of the trained LR model are secret-shared between the parties. Next, the parties run the MPC protocol $\pi_{\mathsf{DP}}$, presented in pseudocode in Prot. 2, to generate noise and add it to the model coefficients to provide DP guarantees. Protocol $\pi_{\mathsf{DP}}$ implements the output perturbation method (or sensitivity method) (Chaudhuri and Monteleoni (2008); Chaudhuri et al. (2011)) in a privacy-preserving way. While the original output perturbation method relies on the fact that the model coefficients are known or disclosed to a single entity, such as a trusted curator, we do not make such an assumption. Instead, the model coefficients remain secret-shared among the computing parties, neither of which knows the true values. The challenge is for the parties to jointly generate noise that is appropriate for the true model coefficients that they cannot see, without learning the true value of the noise. Indeed, no entity should learn the true value of the noise, so that the noisy model coefficients can safely be disclosed at the end of the process (see step 3 in the overview at the beginning of this section), without leaking information that would violate the DP guarantees.

In the output perturbation method, sensitivity is defined using the L2 norm, and the noise vector is sampled from a particular instance of a multidimensional power exponential distribution $h(\eta) \propto e^{-\frac{n\epsilon\Lambda}{2}\|\eta\|}$. Following Sánchez-Manzano et al. (2002), the computing parties can obtain secret shares of a vector $\mathbf{s}$ sampled according to the distribution $h(\eta)$, by following these steps, in which $d$ is the length of the vector (i.e. the number of model coefficients):

1. Generate a $d$-dimensional Gaussian vector $\mathbf{s}$. That is, each coordinate of the vector is sampled from a Gaussian distribution with mean zero and variance one. To this end, Line 1 in Prot. 2 calls $\pi_{\mathsf{GSS}}$ (see pseudocode in Prot. 3) which relies on the transform by Box and Muller (1958) to generate samples of the Gaussian unitary distribution, namely $\lceil d/2 \rceil$ pairs of Gaussian samples. For each pair, on Line 3–4 in Prot. 3, the computing parties securely generate secret shares of two random numbers $u$ and $v$ uniformly distributed in [0,1] by executing $\pi_{\mathsf{GR-RANDOM}}$. In $\pi_{\mathsf{GR-RANDOM}}$, each party generates $l$ random bits, where $l$ is the fractional precision of the power 2 ring representation of real numbers, and then the parties define the bitwise XOR of these $l$ bits as the binary representation of the random number jointly generated. On Line 5–8 in Prot. 3, the parties then jointly compute a secret sharing of $\sqrt{-2\ln(u)}\cdot\cos(2\pi v)$ and of $\sqrt{-2\ln(u)}\cdot\sin(2\pi v)$ using MPC protocols $\pi_{\mathsf{SQRT}}$, $\pi_{\mathsf{SIN}}$, $\pi_{\mathsf{COS}}$, and $\pi_{\mathsf{LN}}$ (Keller (2020)). In case $d$ is odd, one more sample needs to be generated. The parties do so on Line 11–12 in Prot. 3 by executing $\pi_{\mathsf{GSS}}$ to sample a vector of length 2 and only retain the first coordinate.
2. Normalize $\mathbf{s}$, that is divide each coordinate of $\mathbf{s}$ by its L2 norm (Line 2 in Prot. 2). After steps 1-2, the parties have secret-shares of a random $d$-dimensional vector on the unit sphere (this follows from the spherical symmetry of the multivariate Gaussian distribution).

3. In this step the computing parties change the magnitude of the vector obtained above to an appropriate value by sampling the gamma distribution $\Gamma(d, \frac{2}{n\epsilon\Lambda})$ to obtain a value $\gamma$, and multiplying each coordinate of the normalized vector produced in step 2 with $\gamma$. To generate a secret-shared sample $[\![\gamma]\!]$ from the $\Gamma(d, \frac{2}{n\epsilon\Lambda})$ distribution, on Line 3–8 in Prot. 2, the computing parties generate secret shares of $d$ independent samples from the exponential distribution with rate parameter one (here denoted by Exp(1)) and add them. To generate secret shares of one such sample we use the inverse transform sampling over MPC, which consists of computing $-\ln u$, where $u$ is a random number with precision equal to $l$ bits generated by the computing parties within the interval $[0, 1]$:

   (a) On Line 5 the parties execute $\pi_{\mathsf{GR-RANDOM}}$ as in Prot. 3 to generate a random number with precision $l$ in $[0, 1]$. Denote this number by $u$.

   (b) On Line 6 the parties compute secret shares of $-\ln(u)$.

Finally, on Line 9–11 the parties scale the sum by multiplying the secret shares with the factor $\frac{2}{n\epsilon\Lambda}$. On Line 13, they then multiply each coordinate of $\mathbf{s}$ with $\gamma$ to obtain the appropriate magnitude.

The obtained vector is then added to the vector of model coefficients on Line 14.

The importance of protocol $\pi_{\mathsf{DP}}$ stems from the fact that it enables the parties to generate secret shares of noise, without each party learning the true value of the noise that they add to the model coefficients in Line 14 of Prot. 2. The correctness of the protocol follows from the correctness of the inverse transform sampling algorithm, and the fact that $\mathrm{Exp}(1) = \Gamma(1, 1)$ and that $\sum_{i=1}^{d} \Gamma(1, 1) = \Gamma(d, 1)$. Moreover, it follows from the definition of the Gamma distribution that $c \cdot \Gamma(d, 1) = \Gamma(d, c)$. The security of the whole protocol follows from the security guarantees provided by the cryptographic primitives (Keller (2020)).

## 5 RESULTS

**iDASH 2021 Results.** We submitted our approach to a competition hosted by a National Center for Biomedical Computing funded by the NIH. In Track III of the iDASH 2021 competition, participants were invited to submit solutions for learning a ML model from training data hosted by two virtual centers, while providing DP guarantees. The centers represent data owners who have medical records of respectively 831 patients and 882 patients. Both data sets have the same schema, consisting of 1,874 boolean input attributes and a boolean target variable. The goal is to train a classifier for diagnosis of transthyretin amyloid cardiomyopathy using medical claims data (Huda et al. (2021)). Solutions submitted to the competition were required to run on two machines. They were evaluated in terms of (1) training runtime on two nodes with Intel Xeon E3-1280 v5 processors (4 physical cores, hyper-threading enabled) and 64 GiB memory; (2) accuracy on a held-out test of 429 patients.

Tab. 1 contains the results for the best performing teams satisfying the $\epsilon$-DP requirement (with $\epsilon$ set as 3 by the organizers). The first row corresponds to the approach presented in Sec. 4. We implemented the $\pi_{\mathsf{LR}}$ and $\pi_{\mathsf{DP}}$ protocols in MP-SPDZ, an open source framework for MPC (Keller (2020)).[3] Being aware of the pitfalls of implementing DP with floating point arithmetic, our implementation follows the best practice of using fixed-point and integer arithmetic as recommended by, for example, OpenDP.[4] See Appendix C for more details. As the underlying MPC scheme for the iDASH2021 competition, we used semi2k (a semi-honest adaptation of Cramer et al. (2018)) with mixed circuits that employ techniques using secret random bits (extended doubly-authenticated bits; edaBits) (Escudero et al. (2020)). This MPC scheme enables secure 2PC against semi-honest adversaries and complied with the requirements of the competition. As the regularizer for LR training, we used $N(\mathbf{w}) = \frac{1}{2}\mathbf{w} \cdot \mathbf{w}$, in which $\mathbf{w}$ denotes the vector of weights (coefficients) of the LR model, i.e. we used $\Lambda = 1$.

All methods in Tab. 1 provide $\epsilon$-DP guarantees. The differences among the methods are in the utility (accuracy) and in the time taken to train a DP model. Our $\pi_{\mathsf{LR}}$ +$\pi_{\mathsf{DP}}$ approach achieved the highest accuracy of all methods, while taking the longest time to complete. Indeed, the runtime for the $\pi_{\mathsf{LR}}$ +$\pi_{\mathsf{DP}}$ approach is orders of magnitude larger than the runtimes for the other methods. This is because the $\pi_{\mathsf{LR}}$ +$\pi_{\mathsf{DP}}$ approach is the only method in Tab. 1 that uses MPC, while the other methods do not rely on cryptographic techniques. Approach 2 was based on feature selection and training an ensemble of LR models on selected feature subsets, while approach 4 was based on training a decision tree in a DP manner; these approaches were not created by us, and, to the best of our knowledge, their description has not been published in the open literature. In addition to the method from Sec. 4 we

---

[3]See `https://anonymous.4open.science/r/IDASH-MPCheavy-6D69/` for our code.
[4]`https://opendp.org`

Table 1: Results for $\epsilon$-DP with $\epsilon = 3$ and data from two data owners, as provided by the iDASH2021 competition organizers

| | Approach | PETs | Accuracy[3] | Runtime[3] |
|---|---|---|---|---|
| 1. | $\pi_{\mathsf{LR}}+\pi_{\mathsf{DP}}$ (Sec. 4) | MPC & DP | 86.25% | $\sim 15{,}000$ sec |
| 2. | feat. sel. and LR ensemble | DP | 85.31% | 31.942 sec |
| 3. | baseline (Sec. 5) | DP | 84.85% | 0.27 sec |
| 4. | decision tree based | DP | 84.38% | 0.09 sec |

Table 2: 5-fold CV accuracy results for varying number of data owners for $\epsilon$-DP with $\epsilon = 1$.

| # data owners | horizontally distributed | | vertically distributed | |
|---|---|---|---|---|
| | baseline | $\pi_{\mathsf{LR}} +\pi_{\mathsf{DP}}$ | baseline | $\pi_{\mathsf{LR}} +\pi_{\mathsf{DP}}$ |
| 2 | 85.79% | 87.98% | – | 87.98% |
| 4 | 83.36% | 87.98% | – | 87.98% |
| 8 | 76.92% | 87.98% | – | 87.98% |

submitted an MPC-free baseline method to iDASH2021. We describe this method, which corresponds to approach 3 in Tab. 1, below as we also use it for further analysis and comparison in Sec. 5.

**Baseline Method.** The baseline technique follows a FL setup with horizontally distributed data in which each data owner locally trains a model on their own data and adds noise to the model parameters at their end. Each data owner then shares its noisy parameters with a central server who performs averaging of the noisy model parameters and sends the result to the data owners. At the end of this process, each data owner holds the aggregated trained model. In more detail, in the baseline technique, each data owner:

1. Applies L2 normalization to its own instances;
2. Trains a LR model on its normalized instances;[5]
3. Adds noise to the trained LR coefficients as per the output perturbation method (Chaudhuri et al. (2011)).

After going through steps 1-3, the data owners can each publish their perturbed LR coefficients, which we subsequently average to create a final model. Because steps 1–3 provide $\epsilon$-DP (Chaudhuri et al. (2011)), and since the data sets do not have common entries (a case of parallel composition), the overall solution provides $\epsilon$-DP due to the post-processing property of differential privacy.

**Utility on Horizontally and Vertically Distributed Data.** For the results in Tab. 2 we distributed the data evenly among different numbers of data owners, both horizontally and vertically. The baseline technique is only applicable when the data is horizontally distributed, while the $\pi_{\mathsf{LR}} +\pi_{\mathsf{DP}}$ approach works in the vertically distributed scenario as well. Even in the horizontally distributed scenario, the $\pi_{\mathsf{LR}} +\pi_{\mathsf{DP}}$ approach is preferable because it yields a higher accuracy, which becomes even more evident when the data is distributed among multiple data owners. The accuracy of the $\pi_{\mathsf{LR}} +\pi_{\mathsf{DP}}$ approach is independent of the number of data owners and the partitioning of data, as regardless of the partitioning, the computing parties still train a model over all the training data with $\pi_{\mathsf{LR}}$ and subsequently add noise once to the globally trained model coefficients with $\pi_{\mathsf{DP}}$, effectively simulating the global DP paradigm but without the involvement of a trusted curator. The baseline technique on the other hand adheres to the local DP paradigm in which each data owner adds noise to its local model, resulting in more noise in the final aggregated model. Furthermore, the utility of the $\pi_{\mathsf{LR}} +\pi_{\mathsf{DP}}$ approach is independent of the number of instances and/or features owned by each individual data owner, while the accuracy of the baseline technique degrades when individual data owners do not have sufficient instances to train local models that generalize well. This is especially relevant in biomedical applications that are characterized by high-dimensional data sets with relatively few instances.

**Runtime.** As Tab. 3 shows, the number of computing parties, the corruption threshold, and respective MPC schemes do have a substantial effect on the training time. The experiments for Tab. 3 were run with the same training data as in Tab. 1 on co-located F32s V2 Azure virtual machines each of which

---

[5]We used the LR implementation from sklearn for this with penalty='l2' (L2 regularization) and $C = 1$ (the inverse of $\Lambda$).

Table 3: Runtimes of $\pi_{\text{LR}} + \pi_{\text{DP}}$ for different number $r$ of computing parties

| $r$ | Security | Horizontally distributed | Vertically distributed | MPC scheme |
|---|---|---|---|---|
| 2 | Passive | 35687 sec | 38056.92 sec | Cramer et al. (2018) |
| 3 | Passive | 75.83 sec | 454.83 sec | Araki et al. (2016) |
| 3 | Active | 500.28 sec | 1649.07 sec | Dalskov et al. (2021) |
| 4 | Active | 160.50 sec | 838.02 sec | Dalskov et al. (2021) |

contains 32 cores, 64 GiB of memory, and network bandwidth of upto 14 Gb/s. Every computing party ran on a separate VM instance (connected with a Gigabit Ethernet network). The times reported include computing as well as communication times. The training was run for 1000 epochs. with $\epsilon = 1$, $\Lambda = 1$ and with edaBits for mixed circuit computations.

In the horizontally distributed case, the data owners can L2-normalize their instances locally while in the vertically partitioned case the computing parties need to run MPC protocol $\pi_{\text{NORM}}$; this accounts for the difference in runtime between the horizontal and vertical partitioning. As expected, the corruption threshold has the most effect on the run time. Protocols that are secure for an honest majority of players (the protocols presented in Araki et al. (2016), and Dalskov et al. (2021)) are much faster than protocols secure against a dishonest majority (Cramer et al. (2018)). For the same corruption threshold, protocols secure against passive adversaries are faster than protocols secure against active adversaries. The four party protocol proposed in Dalskov et al. (2021) manages to obtain good run times for the case of active adversaries by further reducing the corruption threshold to 25%, i.e. one player out of four can be corrupted by an adversary and the protocol is still secure.

Our results show that MPC implementations for honest majority in the case of realistic sized data sets for genetic studies (a few hundred patients, and a few thousand features) are practical. We can train such models and add DP guarantees on top of MPC in less than 1.3 min for the case of honest majority protocols with passive security. Even in the case of stronger adversarial models, the training can be finished in a few hours, which is still practical for many applications where the increased accuracy payoff is valuable, especially with data that is distributed across multiple owners (Tab. 2).

## 6 CONCLUSION

In this paper, we described a practical and efficient adaptation of distributed logistic regression learning: adding tractable privacy guarantees against model inversion in the absence of a trusted curator which, in real-world scenarios, is often impractical, undesirable, or forbidden. This work led to a 1st place in Track III of the iDASH 2021 Genome Privacy competition. Despite having been formulated for a competition task, the design is intentionally general: it makes no assumption about the data partitioning scenario, number of computing parties / data owners, or the security setting in which it is applied. On the basis of linearity, $\pi_{\text{LR}}$ is interchangeable with all linear learners with no need to reevaluate noise variance. Moreover, the exponential noise mechanism is straightforward to replace by the Gaussian mechanism for scenarios where $(\epsilon, \delta)$-DP guarantees are acceptable. As such, the MPC+DP method can be harnessed for model training across a broad range of use cases without requiring extensive tuning by privacy experts.

The proposed approach effectively offers the advantages of global DP but without the involvement of a trusted curator, because this curator is simulated by an MPC protocol instead. The trade-off between this MPC for global DP approach and the baseline federated method with local DP can be summarized as operating cost (or running time) versus model accuracy. We empirically showed the added utility of collaborative learning with MPC over the standard federated approach. The effect is particularly apparent as the number of disjoint collaborators grows. We also remark that the baseline method, and existing methods that combine MPC with DP in FL, cannot be applied in cases where data is vertically partitioned which is a commonly-found scenario in medicine and advertising. As such, our MPC+DP method allows collaboration in a strictly larger space of applications. Based on performance results, our protocol is extensible to larger data sets while remaining in a realistic time span for model learning, but could be improved further by custom protocol implementations or given the existence of a *correlated randomness dealer* in suitable scenarios. To further improve upon accuracy, a probable research direction is to introduce MPC protocols for feature selection Li et al. (2021) in both horizontal and vertical partitioning schemes.

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

# A  PSEUDOCODE

Pseudocode for $\pi_{\mathsf{LR}}$ is presented in Prot. 4. $\pi_{\mathsf{LR}}$ is based on an existing MPC protocol for training a LR with gradient descent (Keller (2020)), which we extended in two ways to satisfy the conditions:

**(C1)** each input feature vector has an L2 norm of at most 1;

**(C2)** the LR model is trained using L2 regularization.

At the beginning of protocol $\pi_{\mathsf{LR}}$, the computing parties have secret shares of a set of labeled training examples. To satisfy condition **(C1)**, on Line 1–3 the computing parties first apply L2 normalization to the secret shares of each training example by running protocol $\pi_{\mathsf{NORM}}$; pseudocode for $\pi_{\mathsf{NORM}}$ is provided separately in Prot. 1 in the paper.

The computing parties then begin secure training on the privately L2 normalized data from all the data owners. The training begins with initializing the secret shares of the weights (coefficients) of the LR model using Glorot uniform initializer (Glorot and Bengio (2010)). To this end, the computing parties execute protocol $\pi_{\mathsf{INIT}}$ on Line 4. The training is carried out for $n_{iter}$ number of iterations (epochs), which is a public constant agreed upon by all computing parties along with the learning rate $\alpha$, the regularization penalty $\Lambda$, and the momentum $C$. In each epoch, the MP-SPDZ module $\pi_{\mathsf{FWD}}$ for a secure forward pass is called on Line 6, followed by the MP-SPDZ module $\pi_{\mathsf{BKWD}}$ for a backward pass on Line 7. The secret shares of the weights are then updated for every epoch using the MP-SPDZ module for updating the weights. We modified this module to satisfy **(C2)** with L2 regularization as per Line 9 in Prot. 4.

---

**Protocol 4:** $\pi_{\mathsf{LR}}$ for secure logistic regression training

**Input** : A set $S = \{([\![\mathbf{x}]\!], [\![t]\!])\}$ of secret-shared training examples, each consisting of a secret-shared input feature vector $\mathbf{x}$ of length $m$ and a secret shared label $t$; learning rate $\alpha$; regularization penalty $\Lambda$; momentum $C$; number of iterations $n_{iter}$.

**Output** : A secret-shared vector $[\![\mathbf{w}]\!]$ of weights $w_i$ that minimize the sum of squared errors over the training data

1 **for** *training examples* $([\![\mathbf{x}]\!], [\![t]\!])$ *in* $S$ **do**
2 $\quad [\![\mathbf{x}]\!] \leftarrow \pi_{\mathsf{NORM}}([\![\mathbf{x}]\!], m)$
3 **end**
4 $[\![\mathbf{w}]\!] \leftarrow \pi_{\mathsf{INIT}}$                             $\triangleright$ MP-SPDZ module for Glorot uniform initializer
5 **for** $i = 1$ *to* $n_{iter}$ **do**
6 $\quad$ Run $\pi_{\mathsf{FWD}}$                       $\triangleright$ MP-SPDZ module for forward pass
7 $\quad$ Run $\pi_{\mathsf{BKWD}}$                    $\triangleright$ MP-SPDZ module for backward pass
8 $\quad$ Run $\pi_{\mathsf{UPDATE}}$     $\triangleright$ Modified MP-SPDZ module for weight updates with the modified update rule for computing $\Delta\mathbf{w}$: $[\![\Delta\mathbf{w}]\!] \leftarrow C[\![\Delta\mathbf{w}]\!] - \alpha[\![\Delta\mathbf{w}]\!] - \Lambda\alpha[\![\mathbf{w}]\!]$
9 **end**
10 **return** $[\![\mathbf{w}]\!]$

---

# B  ADDITIONAL EXPERIMENTS

## B.1  EFFECT OF PRIVACY BUDGET $\epsilon$ ON ACCURACY OF MODELS TRAINED WITH $\pi_{\mathsf{LR}} + \pi_{\mathsf{DP}}$

Table 4 shows the effect of the privacy budget $\epsilon$ on the accuracy of models trained with the $\pi_{\mathsf{LR}} + \pi_{\mathsf{DP}}$ approach. The accuracy is measured over one of the folds of the train and test data from Sec. 5. The training is done for 1000 epochs and $\Lambda = 1$. The results are as expected, with a larger privacy budget – i.e. less stringent privacy requirements – yielding more accurate models. The observation that the accuracy for $\epsilon = 1$ is at par with the accuracy for $\epsilon = $INF (i.e. when no noise is added) is explained by the fact that adding some noise can positively impact the generalization capability of the model.

## B.2  COMPARISON WITH OTHER PERTURBATION TECHNIQUES

For the $\pi_{\mathsf{LR}} + \pi_{\mathsf{DP}}$ approach (Sec.4) and the baseline technique (Sec. 5), we adopted the sensitivity method that perturbs the model coefficients, i.e. the output perturbation method that was proposed as Algorithm 1 in Chaudhuri et al. (2011). In Table 5 we compare the output perturbation technique with other perturbation techniques, namely objective perturbation and gradient perturbation. For objective perturbation, we ran experiments with Algorithm 2 from Chaudhuri et al. (2011) that adds noise to the

Table 4: Accuracy of models trained with $\pi_{\mathsf{LR}} + \pi_{\mathsf{DP}}$ for different values of $\epsilon$

| $\epsilon$ | ACCURACY |
|---|---|
| 0.001 | 37.72% |
| 0.01 | 50.58% |
| 0.1 | 62.57% |
| 0.5 | 72.51% |
| 1 | 87.42% |
| INF | 86.84% |

objective function itself.[6] For gradient perturbation, we ran experiments with DP-SGD (Abadi et al. (2016)) that adds noise to the gradients.[7] For DP-SGD, we computed the required noise multiplier for given $\epsilon = 1, \delta = 1e - 5$, batch size of 1, 300 epochs, and the number of training examples each data owner holds. This was then passed as an argument to DP-SDG optimizer along with a clipping threshold of 1, learning rate of 0.1, and number of micro batches equal to the batch size.

In the BASELINE-OP method in Table 5, each data owner trains a differentially private LR model locally by perturbing the objective function. The resultant coefficients of the local models are then averaged, resulting in a final DP model. The BASELINE-DPSGD method is entirely similar, but in this method each data owner trains a differentially private LR model by perturbing the gradients learned during training, i.e. with DP-SGD.

As can be seen in Table 5, contrary to what one would expect based on the analysis in Chaudhuri et al. (2011), the accuracy results with this objective function perturbation method were not good on the iDASH2021 data, and far worse than those with the output perturbation method. We attribute this to the high-dimensional nature of the iDASH2021 data (many features and relatively few instances) which is very different from the data sets used for evaluation in Chaudhuri et al. (2011). Similarly, the LR models trained with DP-SGD on the iDASH2021 data are significantly less accurate than those protected with output perturbation.

Table 5: Accuracy results obtained with 5-fold CV for $\epsilon$-DP with $\epsilon = 1$ and 2 data owners

| | APPROACH | ACCURACY | |
|---|---|---|---|
| OUR APPROACH | OUTPUT PERTURBATION | $\pi_{\mathsf{LR}} + \pi_{\mathsf{DP}}$ (SEC. 4) | 87.98% |
| | | BASELINE (SEC. 5) | 85.79% |
| OTHER APPROACHES | OBJECTIVE PERTURBATION | BASELINE-OP | 49.40% |
| | GRADIENT PERTURBATION | BASELINE-DPSGD | 69.77% |

### B.3 COMPARISON WITH OTHER METHODS ON HORIZONTALLY PARTITIONED DATA

We evaluate our MPC+DP approach and compare against existing literature (Pathak et al. (2010) and Jayaraman et al. (2018)) that adopt a combination of PETs to train LR models and provide DP guarantees with the output perturbation technique.[8] The main distinction with our method, is that – similar as in the BASELINE method we adopted in Sec. 5 – these existing approaches let each data owner train a model locally and then add noise to the averaged model parameters using MPC+DP techniques. Because each data owner is required to train a model locally, these existing methods only work in scenarios where the data is horizontally partitioned, unlike our method which is suitable for vertically partitioned scenarios as well. We also note that the amount of noise added by each technique is different.

For the results in Table 6 we distributed the data evenly among different numbers of data owners, in a horizontal manner. We report 5-fold CV accuracy results averaged for 100 runs of noise generation mechanism to consider the randomness in the noise generation. We observe that for 2 data owners, all

---

[6]We implemented this approach using IBM's Diffprivlib library
`https://github.com/IBM/differential-privacy-library`.
[7]We implemented this approach using TF-Privacy
`https://www.tensorflow.org/responsible_ai/privacy/`.
[8]`https://github.com/bargavj/distributedMachineLearning`

the techniques have close performance in terms of accuracy. Similar as for the BASELINE method in Sec. 5, the accuracy of the models trained by existing methods drops with an increase in the number of data owners. This may be because in existing approach, LR models are trained locally by the data owners, while our approach benefits from training an LR model on the combined data and learns a more generalized model. Moreover, our techniques are independent of how the data is distributed among data owners, unlike the methods in Table 6 that work only for horizontally distributed data.

Table 6: Accuracy results for output perturbation obtained with 5-fold CV for $\epsilon$-DP with $\epsilon = 1$ on horizontally partitioned data

| DATA OWNERS | PRIVACY TECHNIQUE | ACCURACY |
|---|---|---|
| 2 | OUR APPROACH ($\pi_{LR} + \pi_{DP}$, SEC. 4) | 87.98% |
| | PATHAK ET AL. (2010) | 86.43% |
| | JAYARAMAN ET AL. (2018) - MPC OUT P | 86.42% |
| 4 | OUR APPROACH ($\pi_{LR} + \pi_{DP}$, SEC. 4) | 87.98% |
| | PATHAK ET AL. (2010) | 85.02% |
| | JAYARAMAN ET AL. (2018) - MPC OUT P | 85.10% |
| 8 | OUR APPROACH ($\pi_{LR} + \pi_{DP}$, SEC. 4) | 87.98% |
| | PATHAK ET AL. (2010) | 84.10% |
| | JAYARAMAN ET AL. (2018) - MPC OUT P | 84.24% |

### B.4 EXPERIMENTS ON OTHER DATA SETS

We further evaluate our approach on the BC-TCGA and GSE2034 data sets of the iDASH 2019 competition.[9] Both data sets contain gene expression data from breast cancer patients which are normal tissue/non-recurrence samples (negative) or breast cancer tissue/recurrence tumor samples (positive) Xie et al. (2016). We perform experiments with a 5-fold CV, where the training data is distributed between 2 data owners in each fold.

**GSE2034** Each instance in this train data set is characterized by 12,634 continuous input attributes and a boolean target variable. There are 895 instances in total. In each iteration of the 5-fold CV, each data owner owns 447-448 instances, 20% of which is held out for testing.

**BC-TCGA** Each instance in this train data set is characterized by 17,814 continuous input attributes and a boolean target variable. There are 1,875 instances in total. In each iteration of the 5-fold CV, each data owner owns 937-938 instances, 20% of which are held out for testing.

The secure training is run for 20 epochs for the BC-TCGA data set and 300 epochs for the GSE2034 data set with $\Lambda = 1$ and $\epsilon = 1$. Table 7 shows accuracy results obtained with a 5-fold CV. To appreciate the inherent difference in difficulty between the GSE2034 and the BC-TCGA classification tasks, as the first row of results in Table 7 we include the accuracies obtained with a model trained in the central learning paradigm, i.e. when all the training data resides with a single data owner, and no noise is added to the model coefficients, i.e. $\epsilon =$ INF. The other rows correspond to the federated setup from Sec. 5 with 2 data owners. The results are in line with the observation from Sec. 5 that the $\pi_{LR} + \pi_{DP}$ approach provides higher utility.

We additionally report the runtime to train the model using $\pi_{LR} + \pi_{DP}$ for these data sets to illustrate the variability in runtimes with respect to the number of training samples, epochs and a number of features in the data set. We see an increase in runtimes for per epoch when compared to the runtimes per epoch on iDASH, which is attributed to a large number of features (about 10x of iDASH2021 for BC-TCGA and 7x for GSE2034). The runtimes for other threat models will follow a similar trend. We see that for larger datasets like these it is still practical to maintain the utility of the model while providing complete privacy guarantees.

### B.5 SCALABILITY OF $\pi_{LR} + \pi_{DP}$ WITH NUMBER OF COMPUTING PARTIES

The number of data holders in our solution is distinct from the number of computing parties. Our solution is general and works with any number of computing servers as well as data holders. In Table

---

[9]http://www.humangenomeprivacy.org/2019/competition-tasks.html

Table 7: Accuracy averaged over 5-fold CV with $\Lambda = 1$

|  | GSE2034 | BC-TCGA |
|---|---|---|
| # INSTANCES $n$ | 895 | 1,875 |
| # FEATURES $d$ | 12,634 | 17,814 |
| CENTRAL LEARNING; 1 DATA OWNER | 65.55% | 98.28% |
| BASELINE (SEC. 5); 2 DATA OWNERS | 51.92% | 91.37% |
| $\pi_{LR}+\pi_{DP}$ (SEC. 4); 2 DATA OWNERS | 64.55% | 95.69% |
| RUNTIME FOR $\pi_{LR}+\pi_{DP}$; PASSIVE 3PC | 276.38 SEC | 57.30 SEC |

Table 8: Runtimes of $\pi_{LR} + \pi_{DP}$ for different number $r$ of computing parties

| MPC SCHEME | $r$ | RUNTIMES | COMM. OVERHEAD |
|---|---|---|---|
| GOYAL ET AL. (2021) (PASSIVE) | 3 | 954.91 SEC | 57922.70 MB |
|  | 4 | 1022.16 SEC | 83667.90 MB |
|  | 5 | 2725.58 SEC | 366679.00 MB |
|  | 7 | 5064.27 SEC | 711226.33 MB |
| CRAMER ET AL. (2000) & CHIDA ET AL. (2018) (ACTIVE) | 3 | 21213.21 SEC | 5247186.89 MB |
|  | 4 | 23244.34 SEC | 7248822.06 MB |
|  | 5 | 68176.24 SEC | 25728263.60 MB |
|  | 7 | 131391.00 SEC | 70080327.38 MB |

8, we report the runtimes and communication overheads to train a LR model with a varying number of computing parties $r$ ranging from 3 to 7. To have a comparison of runtimes and communication overhead for different values of $r$, we use the same MPC scheme for each security setting. The chosen MPC schemes can be used with any value of $r > 2$, and are different from the schemes that we use in Table 3 which were specific and efficient schemes for the given value of $r$. It is due to this use of different schemes that we observe a huge difference in runtimes when compared to the runtimes reported earlier. The schemes in Table 8 are run with secret sharings in $\mathbb{Z}_q$ where $q$ is a prime number[10] and with edaBits for mixed circuit computations.

The training was run on the complete training dataset from iDASH2021 consisting of 1713 training samples and 1874 features for 1000 epochs with GD, $\epsilon = 1$, and $\Lambda = 1$. The runtimes reported include computing as well as communication times. The total amount of data sent by all the computing parties is shown in the last column. The runtimes and the communication overhead increase with an increasing number of computing parties. This is because each party now needs to communicate with a higher number of parties, and the runtimes include communication times. Also, the active security settings take longer runtimes than their passive counterparts for a given $r$. These results are in line with the literature in MPC. The communication overhead in settings with a larger number of computing parties can be reduced with the use of a bulletin board functionality that enables efficient communication among many parties who are simultaneously involved in computations (Agarwal et al. (2019)).

## C  IMPLEMENTATION OF DP IN MPC PROTOCOLS

It is well documented that implementing DP mechanisms using floating-point arithmetic can lead to catastrophic privacy compromises Mironov (2012). The most privacy-conscious choice, taken, for instance, by the OpenDP project[11] is to use fixed-point and integer arithmetic whenever possible. Following this paradigm, in our implementation we have replaced Gaussian and Laplace noise with their discrete approximations. The accuracy of the model is another point where the precision of weights could affect the overall result. Keeping this in mind, we used 32 bits of precision, which is more than sufficient to ensure the correct behavior of the training procedure.

---

[10]Defaults to None in MP-SPDZ and can be a maximum of bit length 256.

[11]https://opendp.org/

We would like to stress that the finite precision issue is inherent to any implementation of DP on a digital computer – it is not specific to our work on DP implemented by MPC protocols. DP theory was created, for the most part, based on continuous distributions. However, all the practical libraries implement DP using finite precision arithmetic. That includes, for example, all the implementations of DP-SGD (which is based on the Gaussian mechanism). It is legitimate to wonder if security guarantees break down in the case when continuous DP mechanisms are implemented on digital computers. However, that question, which has to be asked of all implementations of DP mechanisms based on continuous distributions, is outside the scope of this paper.

