# OpenReview forum: "Cross-Silo Training of Differentially Private Models with  Secure Multiparty Computation"
_ICLR.cc/2023/Conference — Submitted to ICLR 2023_

### Official Review · Reviewer_d2J7 · 2022-10-21

**Confidence:** 4
**Correctness:** 2
**Technical Novelty And Significance:** 2
**Empirical Novelty And Significance:** 2
**Recommendation:** 3

**Clarity, Quality, Novelty And Reproducibility:**

The presentation of the current paper is not clear:
1. It is unclear how to get the update rule in line 8 in Protocol 4. If we choose $C$ and $\Lambda$ to be zero, should it recover the original update rule?
2. There is no formal privacy guarantee of the proposed method.
3. For the baseline methods in Table 1 and Table 5, what are their noise magnitudes to achieve DP? For example, how does the variance of BASELINE-DPSGD in Table 5 scale with the number of samples and computing parties?
4. Why the accuracy of nonprivate method can be worse than the private method with privacy budget 1 in Table 4?
5. It is unclear how to get the hyperparameters of the proposed method.

The main contribution of the proposed method is to use secure computing parties, which enables the training of vertically distributed data. However, such a method seems to be very easy to implement in existing methods and thus allows them to handle the vertically distributed data. Furthermore, what is the advantage of the proposed noise generation method inside MPC compared with existing methods?



**Strength And Weaknesses:**

The strength of the paper:
1. The proposed method enables private training for vertically distributed data, which is an important setting in federated learning.
2. The authors propose a new method for secure sampling of a vector from a Gaussian distribution inside MPC.
3. Empirical results validate the advantage of the proposed method.

The Weaknesses of the paper:
1. It seems that the proposed method requires communication between computing parties at each iteration, which could be the key bottleneck in federated learning. Compared with the output perturbation-based method, which only requires one communication at the end, the proposed method suffers from high communication complexity.
2. There are no theoretical guarantees of the proposed method.
3. The presentation is not clear. For example, the learning problem is not defined. There are lots of undefined notations. More importantly, the random noise magnitudes for several baselines are unclear.
4. It seems to miss an important baseline, i.e., Gu et al. (2021), in the experiments.

**Summary Of The Paper:**

This paper studies the problem of differentially private federated learning. More specifically, in order to improve the utility of the differentially private model in federated learning, the authors propose to combine secure multiparty computation (MPC) with differential privacy (DP) when solving the regularized logistic regression. The main contribution of the paper is a new framework that allows private training for vertically distributed data.

**Summary Of The Review:**

The proposed method seems to be interesting since it can deal with vertically distributed data in federated learning. However, there are some problems that need to be addressed in the current paper (see weaknesses and clarity sections).

---

> ### Author Response · Authors · 2022-11-19
> **On the results in Table 4**
>
> The noise adds randomization to the model parameters. The better performance of private model ($\epsilon=1$) when compared with the model with no output perturbation ($\epsilon=\infty$) is explained by the fact that adding some noise can positively impact the generalization capability of the model.

---

> ### Author Response · Authors · 2022-11-19
> **On the noise magnitudes and hyperparameters**
>
> The sensitivity of the DP mechanism used in our paper is $\frac{2}{n \Lambda}$. Table 1 uses $\epsilon=3$ and Table 5 uses $\epsilon=1$ (cfr. the description of the tables). We will bring clarity on the variance of the DP-SGD method in the camera-ready version. The hyperparameters in our proposed method are the regularization parameter, $\epsilon$ and the precision. We provide analysis on varying $\epsilon$ in Appendix B.1 and other parameters were obtained using trial-and-error to get better utility with 5-fold CV.

---

> ### Author Response · Authors · 2022-11-19
> **On the update rule**
>
> We implemented our proposed idea of MPC+DP in the MP-SPDZ framework which comes with its own module for training LR model and MPC protocol for the update rule. We extended this existing MPC protocol for update rule by adding the regularization parameter. Having the regularization parameter as 0 will result in the original MPC protocol for the update rule in MP-SPDZ.

---

> ### Author Response · Authors · 2022-11-19
> **On other baselines**
>
> Gu et al. propose framework for the horizontally partitioned data and noise is added in-the-clear by each server leading to more noise being added than our proposed approach. We will include this in our baseline results in the camera-ready version.
> The source code for this work was not publicly available at the time of writing the paper.

---

> ### Author Response · Authors · 2022-11-19
> **On the privacy guarantees**
>
> Our protocols were designed and implemented using MP-SPDZ, which follows the SPDZ framework. The privacy of the MPC protocols follows immediately from the privacy guarantees and proofs of the original SPDZ framework, as pointed out in the paper. The DP guarantees follow from the analysis presented in [Chaudhuri, Kamalika et al.].
>
> Chaudhuri, Kamalika, Claire Monteleoni, and Anand D. Sarwate. "Differentially private empirical risk minimization." Journal of Machine Learning Research 12.3 (2011).

---

> ### Author Response · Authors · 2022-11-19
> **On the proposed approach**
>
> Thank you for your time and effort in reviewing our work. Please find our response to your questions below. We appreciate it if you could take these into account when updating your score!
>
> The key idea behind our proposed framework is to distribute the aggregator in a typical federated learning setting. In place of a single aggregator, we have a few servers connected by pairwise private authenticated channels. In our scenario, the data-holders (typically called clients in a federated learning setting) only participate in the computation during a preliminary phase when they secret share their data with the servers. This is a non-interactive procedure. We want to emphasize that there is no interaction or computation among clients/data holders  in our solution.
>
> The servers indeed run a highly interactive protocol. However, in a typical scenario, while clients can potentially run in the millions, these servers are a few (typically 3, 4 servers) and are connected by high speed networks. Thus, the higher number of rounds in our solution is not a concern. The theoretical privacy guarantees of our protocol are well stated in the paper. Privacy during the computation follows from the security of the underlying MPC protocols. Privacy of the output follows from the differential privacy mechanisms we use.
>
> We tried our best to introduce the notations in Section 2 and cite the relevant works. We propose a general idea to train ML models in federated scenario. The specific learning problem that we address in our experiments is the learning task defined in iDASH 2021 competition (cfr. Sec 5 iDASH 2021 results paragraph 1) where the goal is to train a classifier for diagnosis of transthyretin amyloid cardiomyopathy using medical claims data, and we propose to train a logistic regression model. We will make the learning problem more clear in the Introduction in the camera-ready version.

---

### Official Review · Reviewer_vZqr · 2022-10-24

**Confidence:** 4
**Correctness:** 4
**Technical Novelty And Significance:** 2
**Empirical Novelty And Significance:** 3
**Recommendation:** 6

**Clarity, Quality, Novelty And Reproducibility:**

quality: the paper has good technical and empirical quality.
clarity: the paper is well-written.
originality: the paper lacks a bit of methodological novelty, but the fact that it got the different components together working well together is new.
reproducibility: the results are likely reproducible, given the authors attach a link to their code.

**Strength And Weaknesses:**

strength: the paper is well-written and the empirical results are solid.

weaknesses:
- the main method in the paper is a combination of existing protocols in MPC and differentially private statistics release, and thus the paper lacks methodological novelty. but this perhaps isn't a super serious issue, given that authors demonstrated their point and showed good performance.
- the paper only considers a single dataset / benchmark on which results are reported. this makes understanding the robustness of the method and results a bit difficult.
- the method is likely so far most practical for small (to perhaps moderate) size problems, given the large compute overhead.

**Summary Of The Paper:**

the paper studies a method which combines MPC with DP that bypasses utility challenges of federated learning with local DP (with small datasets). authors show that the method achieves good performance by reporting their submission entry being 1st place for an iDASH 2021 challenge, at the cost of substantially more compute time.

**Summary Of The Review:**

the paper studies a method which combines MPC with DP that bypasses utility challenges of federated learning with local DP (with small datasets). authors show that the method achieves good performance by reporting their submission entry being 1st place for an iDASH 2021 challenge, at the cost of substantially more compute time. the paper is well-written, is of high quality, but lacks a little in methodological novelty.

---

> ### Author Response · Authors · 2022-11-19
> **On the additional experiments**
>
> We demonstrate the robustness and effectiveness of our proposed approach against two other high dimensional datasets (please see Appendix B.4) and compare with other techniques (please see Appendix B.2 and B.3) in the Appendix.

---

> ### Author Response · Authors · 2022-11-19
> **On the practical use of our proposed approach**
>
> We performed experiments on realistic sized data sets for genetic studies (a few hundred patients, and a few thousand features) which is practical. Even in the case of stronger adversarial models, the training can be finished in a few hours, which is still practical for many applications where the increased accuracy payoff is valuable, especially with data that is distributed across multiple owners.

---

> ### Author Response · Authors · 2022-11-19
> **On the methodological novelty**
>
> Thank you for your time and effort in reviewing our work, and for calling out that our paper is of high quality! We spent a great amount of time and effort in developing this approach, which is novel and  practical. We appreciate it if you could take these into account when updating your score!
>
> The main contributions of our paper are on how best we combine the existing primitive techniques in MPC and DP to develop machine learning training techniques for the federated scenario. Doing so, we propose a novel way of training ML models that (i) apply for any partitioning of the data; (ii) emulates global DP paradigm to preserve utility; (iii) is independant of the number of data holders and does not require any communication; and (iv) the fact that our idea is general which can be extended to any ML models with any DP mechanisms providing MPC-as-a-Service model. We recall that no existing work achieves the above.

---

### Official Review · Reviewer_DC85 · 2022-10-24

**Confidence:** 2
**Correctness:** 4
**Technical Novelty And Significance:** 3
**Empirical Novelty And Significance:** 4
**Recommendation:** 6

**Clarity, Quality, Novelty And Reproducibility:**

# Clarity

The paper is very well written and clear.

# Quality

I think the paper and its appendices contain material for a good paper, up to some restructuring and maybe adding a few more baselines (cf above)

# Novelty

The contributions of the paper, although limited in scope, are novel to the best of my knowledge

# Reproducibility

Code is provided by the authors. I have not checked it exactly to understand how easy it would be to reproduce it.

**Strength And Weaknesses:**

# Strengths

1. The paper is very well written, which was not a given given the very technical nature of the topics tackled (MPC *and* DP). I really thank the authors for their writing efforts
2. The idea proposed is general enough to be applied to any model amenable to the output perturbation method of Chaudhuri et al., which includes e.g. SVMs
3. The method seems to reach the best performance on the current benchmark
4. The method should yield results equivalent to centralized training (up to discretization issues), which is great for healthcare applications where there is high heterogeneity and little data per center

# Weaknesses

5. The paper is still limited to the output perturbation method, and only covers the logistic regression
  - In particular, I think the abstract is misleading and should reflect more the focus on logistic regression
6. The structure of the paper is unbalanced between the main paper and the supplementary material. The supplementary material addresses many points that are lacking in the main paper, including extension to other datasets, more interesting baselines, issues with finite precision. Several appendices are not even referred to in the main paper.
7. The choice of baselines (even after the extension in the appendix) is not entirely fair:
  - DP-SGD is performed with a batch size of 1, which is a worst-case approach.
  - Further, DP-SGD is only used with local trainings followed by ensembling. The authors do not consider a FedAvg + DP-SGD approach which would be natural in this setting. Such an approach was already used e.g. in the iDASH 2020 competition (Beguier et al. Differentially Private Federated Learning for Cancer Prediction)
  - For the method of Jayaraman et al. 2018 in table 6, the variant with output perturbation is used, although this paper shows that the gradient perturbation yields better results
8. Regarding the experimental setup :
  - It would be better to add error bars (where applicable) to understand the significance of the results. In particular, this could be possible for the cross-validation results of Table 2 and Table 6
9. For vertical FL, one of the main challenges is to align record across parties. Which method do the authors use to tackle this question?


Minor comments:
- How do you explain the fact that in Table 7, central learning yields a better performance than the proposed MPC method? Is it due to finite precision?
- how much does the initial secret sharing transfer (encrypted data points) weigh with respect to the total communication cost?
- For L2 data normalization, for horizontal FL, it is probably simpler to apply it prior to creating shared secrets. When computing the total runtime, is it contained in it?

**Summary Of The Paper:**

This paper investigates the problem of training a logistic regression (LR) model with differential privacy over distributed data. In order to improve over local DP approaches, it relies on multiparty computation to train a pooled-equivalent model on secret shares, adding differential privacy on the final model thanks to the output perturbation method of Chaudhuri et al. Experiments are performed on several datasets and demonstrate the improved performance, at the expense of a large communication cost (which can be mitigated depending on the protocol used and the threat model)

To sum up, the main contributions of the paper are:
1) Proposing an MPC protocol to run the output perturbation of Chaudhuri et al. for logistic regression, working for both horizontal and vertical FL
2) Extending MP-SPDZ for logistic regression training with L2 regularization
3) Empirical study of the benefits of the proposed approach in terms of performance (for a given privacy budget) and communication, mainly on the iDASH 2021 track III competition dataset but also on TCGA and GSE2034 in the supplementary

**Summary Of The Review:**

This paper proposes a novel protocol for logistic regression with differential privacy in a distributed setting, which seems to reach good results. Although I think the paper would benefit from a restructuring and adding some baselines, I am bending in favor of acceptance, hence the current score.

---

> ### Author Response · Authors · 2022-11-19
> **On visual representation of results and referring to experiments in the appendix**
>
> Thank you for the suggestion! This indeed is a welcoming suggestion. Due to the page-limit, we could not add such graphs and also had to move many of the relevant experiments to the appendix. We will try to include error graphs in the camera-ready version and refer the appendices in the main content.

---

> ### Author Response · Authors · 2022-11-19
> **On vertically partitioned data**
>
> The main focus of our paper is on training ML models in federated setup with different data partitioning scenarios. In our work, we assume that the record alignment for vertical or mixed partitioning is already done using privacy preserving techniques as in [Mohassel, Payman et. al.] prior to the start of the training. We will mention such techniques and how they can be used in combination with our approach in the camera-ready version.
>
> Mohassel, Payman, Peter Rindal, and Mike Rosulek. "Fast database joins and PSI for secret shared data." Proceedings of the 2020 ACM SIGSAC Conference on Computer and Communications Security. 2020.

---

> ### Author Response · Authors · 2022-11-19
> **On minor comments**
>
> 1. The centralized setup does not provide DP guarantees and so should result in the better performance than all approaches. Our approach with MPC, apart from providing input privacy and preserving utility as in the centralized learning, also provides output privacy. We do so by perturbing the coefficients of the LR model i.e. by adding Laplacian-based noise which can effect the utility of the model, i.e. decreased utility when compared to central learning. Our proposed approach with MPC and emulation of global DP achieves the utility as close to the centralized setup while providing both input and output privacy.
>
> 2.  The overhead of initial secret-sharing depends on the number of values to be secret shared and the number of computing parties (in general). This in addition also depends on the network setup (e.g. WAN,LAN) between the data holders and the computing parties.
> Assuming that all data holders parallely secret-share their data, each data holder, in our case, has a communication overhead of ~216 bytes when secret sharing data with 3 computing servers which is negligible compared to the communication overhead of running MPC protocols for training. We will include this analysis in the camera-ready version.
>
> 3. We do note that in the horizontally distributed case, the data owners can L2-normalize their instances locally (cfr. Sec. 5 Runtime paragraph 2). The times reported in Table 3 do not include the time for locally normalizing the training instances.

---

> ### Author Response · Authors · 2022-11-19
> **On the choice of baselines**
>
> We welcome the reviewer's suggestions on including more relevant baselines. We want to clarify that we ran experiments with batch size = 32 and micro batch size of 1 in the DP-SGD algorithm. We will correct this in the camera-ready version and include additional results with DP-SGD. We will also explore the DP-SGD mechanism designed for Federated Setting and include our analysis in the camera-ready version.
>
> We want to call out that our approach is suitable for different data partitioning scenarios in the federated setup and achieves utility close to the centralized paradigm as opposed to the traditional federated setups mentioned above that are applicable for the horizontally partitioned data.

---

> ### Author Response · Authors · 2022-11-19
> **On the choice of perturbation technique**
>
> Thank you for your time and effort in reviewing our paper, and for calling out our work novel.  We are glad that you found our paper as well-written and that our approach is a great choice for healthcare applications. We thank you for your comments and we spent a great amount of time and effort in developing this approach. We appreciate it if you could take our reply into account when updating your score!
>
> Our proposed approach to combine MPC and DP is fairly general and can be extended to other ML models and DP mechanisms. Based on our experiments in Appendix B.2, the output perturbation method gave better performance than others on the iDASH2021 data (high-dimensional data i.e. many features and relatively few instances). As a starting point for our proposed general idea,  we designed specific MPC protocols for output perturbation for the task in iDASH 2021. We note that our proposed idea can be extended to other perturbation techniques with design of relevant MPC protocols.

---

> ### Comment · Reviewer_DC85 · 2022-12-04
> **Thanks for your answer**
>
> I thank the authors for their answer to my comments. Regarding the perturbation technique, I agree that the idea of using MPC + DP is fairly general, but as the implementation shown in the paper is limited to the logistic regression, which limits the contribution. I will therefore keep my score as is.

---

### Official Review · Reviewer_QAdF · 2022-10-25

**Confidence:** 4
**Correctness:** 2
**Technical Novelty And Significance:** 1
**Empirical Novelty And Significance:** Not applicable
**Recommendation:** 3

**Clarity, Quality, Novelty And Reproducibility:**

Clarity: The authors explain the method extremely well.
Quality:
- DP mechanism needs to be explained more thoroughly. For example: (1) why not use discrete Gaussian [1] (2) Does your method provide DP guarantees even if it is using Box-Muller on not real values that have been shown to be vulnerable to floating point attacks (3) does the method provide epsilon or epsilon, delta guarantees.
- how do parties match their features when data is vertically split? would not they need to run an MPC protocol for that as well?
- what is the baseline of an approach without DP and MPC and use of fixed point arithmetic? Please also consider reporting communication overhead.
Novelty:
There is a bit of work in this space and it seems that this paper combines existing techniques. Is it the performance that is the main contribution? If so how does it compare with other works (SecureML, [2]) or with trusted-execution based works?

Related work:
[1] The Discrete Gaussian for Differential Privacy by Canonne et al.
[2] CaPC Learning: Confidential and Private Collaborative Learning by Choquette-Choo et al.
SecureML by Mohassel and Zhang.


**Strength And Weaknesses:**

Strengths:
- a really well-written paper, extremely easy to follow
- the problem of cross-silo learning is addressed completely by considering security, privacy and integrity
- related work is very well-explained (recommendations on expanding are below)

Weaknesses:
- the novelty of the work compared to existing MPC+DP approaches is not clear
- DP guarantees need further investigation (e.g., why delta is missing and why not use Gaussian discrete noise)
- the method seem to work only on LR models
- baselines method (no DP and no MPC is missing)
- comparison with other MPC+DP approaches is missing
- how parties combine their data in vertical setting is not explained

**Summary Of The Paper:**

The paper considers the problem of secure and privacy-preserving distributed ML training (specifically logistic regression model). It does so by combining MPC and DP. The approach by normalizing the data and then adding the noise to the parameters proportional to sensitivity of how this data would change the parameters. The protocol is completely distributed and does not require a trusted party.
The paper seems to describe the method that was used in iDash competition last year.

**Summary Of The Review:**

This is a really nicely written paper. However contributions seem minor theoretically. Many details that need to be in main body (e.g., other datasets, generalisability, communication overhead appear in appendix) and it is hard to determine what is the generalizability of the methods.

---

> ### Author Response · Authors · 2022-11-19
> **On the baseline approach**
>
> The accuracy results with floating-point (in-the-clear i.e.~without MPC and DP) is same as the accuracy with fixed-point (in-the-clear) when using 32 bit precision. We will include this in the camera-ready version.
>
> Thank you for the suggestion! We will include communication overheads for all our experiments in the camera-ready version. We want to note that we do report communication overhead along with runtimes when evaluating our proposed approach for scalability in Appendix B.5.

---

> ### Author Response · Authors · 2022-11-19
> **On vertically partitioned data**
>
> The main focus of our paper is training ML models in federated setup with different data partitioning scenarios. In our work, we assume that the record alignment for vertical or mixed partitioning is already done using privacy preserving techniques as in [Mohassel, Payman et. al.] prior to the start of the training. We will mention such techniques and how they can be used in combination with our approach in the camera-ready version.
>
> Mohassel, Payman, Peter Rindal, and Mike Rosulek. "Fast database joins and PSI for secret shared data." Proceedings of the 2020 ACM SIGSAC Conference on Computer and Communications Security. 2020.

---

> ### Author Response · Authors · 2022-11-19
> **On DP mechanism**
>
> Our proposed method adapts $\epsilon$-DP privacy model based on [Chaudhuri, Kamalika et. al.](cfr. Sec. 4 paragraph 2). Due to this, we propose MPC protocols to generate Laplacian-based noise to provide $\epsilon$-DP guarantees and do not require to use Gaussian mechanism. Our idea to emulate the global DP paradigm can be extended to provide $(\epsilon,\delta)$-DP guarantees by designing appropriate MPC protocols.
>
> We provided a brief explanation on implementation of DP mechanisms in MPC in Appendix C.
> We are aware that implementing DP mechanisms using floating-point arithmetic can lead to privacy issues. We follow the best practices for implementing DP mechanism using fixed-point or integer arithmetic (https://opendp.org/) and replace the Laplace-based noise with its discrete approximation. The validity of our approach (e.g. $\pi_{\mathsf{GSS}}$ that relies on discrete version of Box-Muller transform) is the same as any other implementation of DP using fixed precision (as recommended by OpenDP). We used 32 bits of precision, which is more than sufficient to ensure the correct behavior of the training procedure [Mironov, Ilya].
>
>
>
>
> Chaudhuri, Kamalika, Claire Monteleoni, and Anand D. Sarwate. "Differentially private empirical risk minimization." Journal of Machine Learning Research 12.3 (2011).
>
> Mironov, Ilya. "On significance of the least significant bits for differential privacy." Proceedings of the 2012 ACM conference on Computer and communications security. 2012.

---

> ### Author Response · Authors · 2022-11-19
> **On the choice of LR model and generalizability**
>
> When working with MPC it is important to keep algorithms lightweight. What works best in-the-clear, i.e. without encryption, is not necessarily what works best on top of MPC. Generalized linear models, which are a method of choice for many practical ML applications, are an attractive option for privacy-preserving ML, because the MPC protocols for training these models are substantially more efficient than e.g. those for ANNs. We use a simple yet powerful idea: to replace the trusted aggregator from the global DP paradigm by computing servers in MPC. This general idea can be applied to train all kinds of ML models with different DP mechanisms by designing the appropriate MPC protocols which, as a starting point, we did for logistic regression with output perturbation that provides $\epsilon$-DP guarantees.

---

> ### Author Response · Authors · 2022-11-19
> **On novelty**
>
> Thank you for your time and effort in reviewing our work. Please find our response to your questions below. We appreciate it if you could take these into account when updating your score!
>
> The main contribution of our paper relies on a simple yet powerful idea: to use multiple computing parties as ''aggregators'' in an FL setting. By having these multiple aggregators running MPC protocols based on secret sharing, we achieve several properties: (i) The approach immediately works with data partitioned in an arbitrary way, i.e. horizontally, vertically, or otherwise; (ii) We can emulate a trusted dealer in the global DP scenario, thus keeping the noise needed for obtaining DP guarantees to the minimum necessary, and preserving good utility; (iii) We avoid any communication among data holders. Most of the previous proposals that combine MPC+DP do not generate noise fully within the MPC protocol. Most of such works assume that the noise is generated by the data holders themselves and such methods are susceptible to misbehavior by the data holders. Our method ensures that security against malicious parties can be achieved in a straightforward way. We argue that such solution is desirable in many scenarios.
>
> The fact that our approach works for vertically partitioned data while previous approaches can only handle horizontally distributed data is a direct way to tell that our approach is different from existing work, and that the manner in which we combine MPC and DP is a useful and effective novelty in itself. We compare our work with some of the relevant and related works in the Appendix (cfr. Appendix B.2 and B.3)
>
> We note that SecureML does provide input privacy but fails to provide output privacy as we do. The CaPC does not work for the vertically partitioned data scenario.

---

### Decision · Program_Chairs · 2023-01-20

**Decision:**

Reject

**Justification For Why Not Higher Score:**

The main concern about this paper is the novelty of the algorithm.

**Justification For Why Not Lower Score:**

N/A

**Metareview: Summary, Strengths And Weaknesses:**

This paper proposes an MPC solution for training differentially private models, which achieves higher accuracy than a pure DP approach while providing the same formal privacy guarantees.

Strengths:

+The paper is well written

Weaknesses:

-The novelty is limited because MPC has been used for training differential private models in many prior works

-Lack of thorough experimental evaluation

The main concern about this paper is the novelty of the algorithm. Even after the author's response, it does not gather sufficient support.